# Single-cell analysis of *Schistosoma mansoni* identifies a conserved genetic program controlling germline stem cell fate

Pengyang Li[1], Dania Nanes Sarfati [2,6], Yuan Xue[1,6], Xi Yu[1], Alexander J. Tarashansky[1], Stephen R. Quake [1,3,4] & Bo Wang [1,5 ✉]

Schistosomes are parasitic flatworms causing one of the most prevalent infectious diseases from which millions of people are currently suffering. These parasites have high fecundity and their eggs are both the transmissible agents and the cause of the infection-associated pathology. Given its biomedical significance, the schistosome germline has been a research focus for more than a century. Nonetheless, molecular mechanisms that regulate its development are only now being understood. In particular, it is unknown what balances the fate of germline stem cells (GSCs) in producing daughter stem cells through mitotic divisions versus gametes through meiosis. Here, we perform single-cell RNA sequencing on juvenile schistosomes and capture GSCs during de novo gonadal development. We identify a genetic program that controls the proliferation and differentiation of GSCs. This program centers around onecut, a homeobox transcription factor, and boule, an mRNA binding protein. Their expressions are mutually dependent in the schistosome male germline, and knocking down either of them causes over-proliferation of GSCs and blocks germ cell differentiation. We further show that this germline-specific regulatory program is conserved in the planarian, schistosome's free-living evolutionary cousin, but the function of onecut has changed during evolution to support GSC maintenance.

[1] Department of Bioengineering, Stanford University, Stanford, CA 94305, USA. [2] Department of Biology, Stanford University, Stanford, CA 94305, USA. [3] Department of Applied Physics, Stanford University, Stanford, CA 94305, USA. [4] Chan Zuckerberg Biohub, San Francisco, CA 94158, USA. [5] Department of Developmental Biology, Stanford University School of Medicine, Stanford, CA 94305, USA. [6]These authors contributed equally: Dania Nanes Sarfati, Yuan Xue. ✉email: wangbo@stanford.edu

Schistosomes are parasitic flatworms that cause one of the most prevalent but neglected infectious diseases, schistosomiasis[1]. With over 250 million people infected worldwide and a further 800 million at risk of infection, schistosomiasis imposes a global socioeconomic burden comparable to that of tuberculosis, HIV/AIDS, and malaria[1–3]. The disease transmission requires the passage of parasites through two hosts during its life cycle, a molluscan intermediate host and a mammalian definitive host (e.g., human). This complex life cycle requires the deployment of multiple specialized body plans in order to infect and reproduce within each host. These transitions are enabled by a population of stem cells that undergo several waves of proliferation and differentiation[4–6].

The schistosome life cycle begins with the parasite egg being excreted from the mammalian host into freshwater. It hatches into a free-swimming larva serving as a vehicle for approximately a dozen stem cells and transforms into a sporocyst once inside a snail host[4,5,7,8]. Stem cells in the sporocyst can undergo either self-renewal or enter embryogenesis, producing more sporocysts or massive numbers of infectious progeny (called cercariae)[4,5,8–10]. Cercariae then emerge from the snail back into the water, burrow through the skin of a mammalian host, migrate to species-specific niches in the host vasculature, and transform into juvenile parasites. This transition initiates the sexual portion of the parasite's life cycle. Stem cells packed in cercariae proliferate to initiate the growth of juveniles and eventually build sexual reproductive organs de novo, through a developmental program that appears to be modulated by host hormonal and immune cues[5,11,12]. Sexually mature male and female worms mate, and each pair can produce hundreds of fertilized eggs daily, which are excreted to initiate the next life cycle. It is worth noting that the eggs, not the worms, cause pathology as they become trapped in the host tissues and induce extensive granulomatous inflammation[13].

We recently conducted a single-cell transcriptomic study comparing stem cells from Schistosoma mansoni asexual (intramolluscan) and sexual (intramammalian) stages[5]. Among the stem cells that are carried by cercariae and transmitted from snail to mammal, only a subset of them commits to the germline fate. These primordial germ cells initially express a schistosome-specific factor, eledh (Sm-eled, for brevity the prefix "Sm" will be omitted from gene names in the rest of the paper wherever species origin is unambiguous), and during germline specification activate one of the schistosome homologs of nanos (nanos-1), an RNA binding protein conserved across metazoans[5,14–18]. Unlike most animals that segregate their germline from soma during embryonic development[15,19,20], schistosomes specify their germ cells from a somatic cell lineage at the onset of juvenile development.

While our previous work has identified the origin of the schistosome germline, it remains unclear which molecular regulatory program(s) underlies the separation of somatic and germ cell lineages[5]. In contrast to their distinct cellular fates, somatic and germline stem cells share strikingly similar gene expression signatures, including genes involved in regulating the cell cycle, a set of conserved RNA binding proteins that are often associated with stem cell multipotency[4–6], and transcription factors such as nuclear factor Y subunits[18]. Disrupting these key regulators leads to defects in stem cell maintenance and proliferation in both somatic and germline lineages[4,18]. The molecular distinction between somatic and germline stem cells is thought to be similarly blurry in other flatworms, such as schistosome's free-living evolutionary cousin, the planarian[20,21]. Planarian germline stem cells resemble their somatic stem cells both in terms of morphology and molecular signatures, and can even contribute to somatic tissues during tissue regeneration after injury[20–24].

In this study, we perform single-cell RNA sequencing (scRNAseq) on juvenile S. mansoni to characterize the germline stem cells (GSCs). We focus on this life-cycle stage because juvenile schistosomes contain abundant stem cells, including cells in the transition between somatic and germline fates[5]. We succeed in capturing the GSCs and identify their transcriptional signatures using the self-assembling manifolds (SAM) algorithm[25], a method that excels in identifying subtle transcriptional differences between otherwise similar groups of cells. Through RNA interference (RNAi) mediated gene knockdown, we evaluate the function of GSC-enriched transcripts and find that a schistosome homolog of onecut homeobox transcription factor, onecut-1, is the key regulator that balances the proliferation and differentiation of GSCs (in particular, male GSCs, or spermatogonial stem cells). onecut-1 functions through complex epistatic interactions with several other genes, including eled, nanos, and boule. As nanos and boule are broadly conserved germline developmental regulators[14–18,26–31], and onecut expression has been detected previously in mammalian testes[32–35], we posit that this gene set may represent a conserved germline specific regulatory program. Consistent with this idea, we find that onecut is indeed essential in the male germline of planarian Schmidtea mediterranea, though its specific function has changed. While onecut protein family has been extensively studied across animals (e.g., fly and mouse)[33–38], its role in regulating germline development has been previously unknown.

## Results

**scRNAseq identifies GSC-specific transcriptional signatures.** We have previously performed scRNAseq on flow-sorted $G_2/M$ phase cells to enrich for stem cells that are actively cycling[5,25]. However, this approach failed to capture nanos-1 expressing GSCs. To overcome this limitation, we here sequenced cells from all tissues in juvenile schistosomes using Smart-seq2[39–41]. We collected 10945 single-cell transcriptomes, of which 7657 passed the initial quality control (see "Methods" section).

Figure 1a shows that SAM analysis divided these cells into several major clusters, but only one population expresses ago2-1, an argonaute homolog that is ubiquitously and specifically expressed in all schistosome stem cells[4–6,25]. Performing SAM analysis on ago2-1$^+$ cells (1464 cells) identified nine clusters (Fig. 1b), four of which correspond to stem cell subtypes we reported earlier: $\mu$, $\delta'$, $\varepsilon_\alpha$, and $\varepsilon_\beta$[5,25]. The five newly found populations include one that contains nanos-1 expressing cells. The other four contain cells expressing genes often associated with differentiated tissues (Supplementary Data 1), for example, troponin (a common muscle gene[25,42]), complexin (cpx, a neural gene), cathepsin B endopeptidase (cb2, a gene expressed in intestinal and parenchymal cells[43,44]), and tetraspanin-2 (tsp-2) co-expressed with Sm25 (both are known markers for schistosome epidermal lineage[45]). Since these populations still express common stem cell markers (e.g., ago2-1, cyclin B, pcna, polo-like kinase, histone h2a, and h2b) and are actively dividing (Supplementary Fig. 1), they are likely tissue type-specific progenitors, though further functional validation would be required to prove their respective identities and fates.

Hierarchical clustering of the cell populations (see "Methods" section) divided them into two major groups, one grouping $\mu$-, $\delta'$-, $\varepsilon_\alpha$-, and $\varepsilon_\beta$-cells together with nanos-1$^+$ cells, and the other containing the four putative progenitor populations (Fig. 1c). Unlike progenitor populations that express a large number of population-enriched genes, $\mu$-, $\delta'$-, $\varepsilon_\alpha$-, $\varepsilon_\beta$-cells and nanos-1$^+$ cluster only express a small number of population-enriched genes beyond the common stem cell markers (all population-enriched genes are provided in Supplementary Data 1). Instead, this group

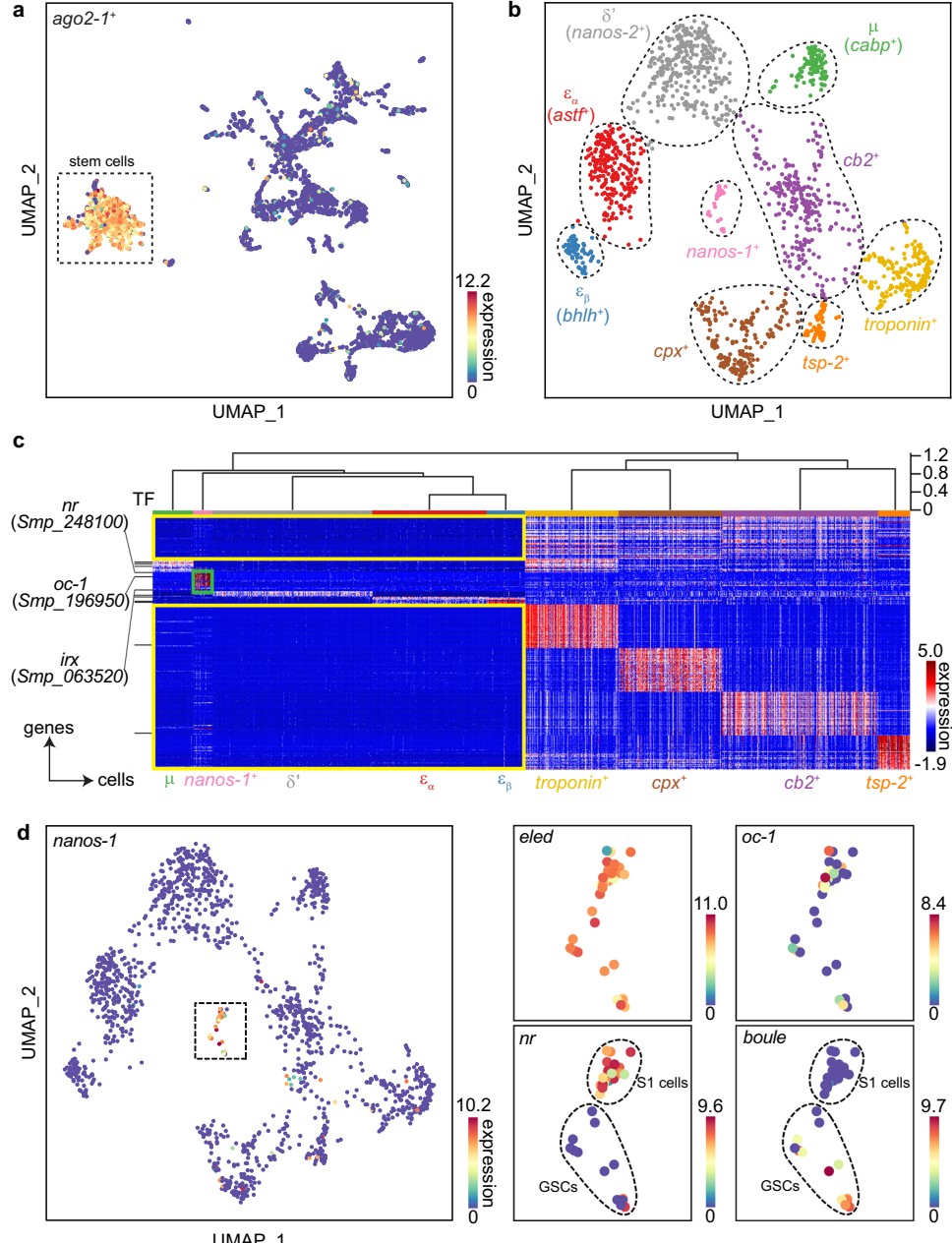

**Fig. 1 scRNA-seq identifies schistosome stem cell subpopulations and GSC transcriptional signatures.** UMAP projections showing scRNA-seq data of **a** all cells and **b** *ago2-1*[+] stem cells (boxed in **a**) only. The manifolds are reconstructed by SAM[25]. In **a**, *ago2-1* expression is overlaid. In **b**, cells are color-coded by Leiden clusters and annotated based on marker gene expression. **c** Heatmap showing the expression of genes enriched in specific stem cell populations, with the mean expression of each gene shifted to 0. Top: the populations are grouped through hierarchical clustering in the PC space, with the height of the branches representing correlation distance between populations. Left: annotated transcription factors (GO: 0003677 DNA binding and GO: 0003700 DNA-binding transcription factor activity) are indicated by black lines. Yellow boxes: genes that lack expression in the presumptive multipotent group compared to the progenitor group. Green box: genes enriched in *nanos-1*[+] population. For the sake of visualization, we only show the top 50 markers per cluster. The complete list of markers is provided in Supplementary Data 1. **d** UMAP projections showing the expression of *nanos-1* in stem cells (left) and the expression of *eled* (a previously characterized GSC marker[5]), *oc-1*, *nr*, and *boule* in *nanos-1*[+] cells (right). *nr* and *boule* expression separate *nanos-1*[+] cells into S1 stem cells and GSCs.

is separated from the progenitor group primarily based on the lack of marker gene expression (yellow boxes in Fig. 1c), consistent with the idea that multipotent stem cells repress genes that are involved in cell differentiation[46].

Focusing on the *nanos-1*[+] cells, we examined the expression pattern of their enriched genes (green box in Fig. 1c). This list includes transcription factor *onecut-1* (*oc-1*, Smp_196950, Supplementary Fig. 2), which is expressed throughout this

population (Fig. 1d). We also identified two clearly distinguishable subclusters marked by *boule* (Smp_144860) and a nuclear receptor (*nr*, Smp_248100), respectively. As *nr* is known to be expressed in the *nanos-1*[+] stem cells (also known as S1 cells) in primordial vitellaria[47,48], a yolk cell-producing somatic reproductive organ in female schistosomes, these observations suggest that the *nanos-1*[+] cluster contains two transcriptionally similar but distinct cell types: S1 cells (*nr*[+]) and GSCs (*boule*[+]).

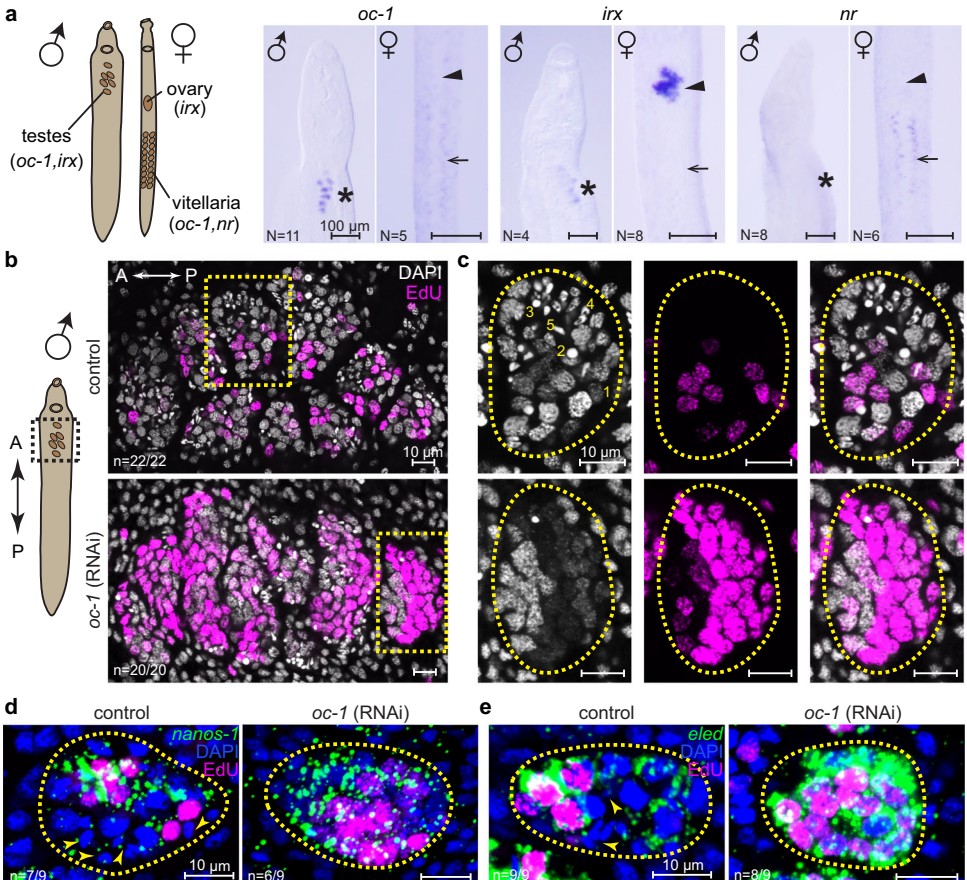

**Fig. 2 *oc-1* RNAi leads to over-proliferation of male GSCs. a** WISH images of *oc-1*, *irx*, and *nr* expression in male and female juvenile parasites. Left: schematic showing the locations of major reproductive organs in schistosome juveniles. Asterisks: primordial testes; arrowheads: ovary primordia; arrows: vitellaria primordia. *N*: number of samples showing similar results over two independent experiments, each using parasites collected from a separate infection. **b** Confocal images of DAPI/EdU-stained testes in control and *oc-1* RNAi male juvenile parasites. The imaged areas correspond to the dashed box in the schematic on the left. A: anterior; P: posterior. **c** Magnified views of individual testis lobules corresponding to yellow boxes in **b**. Dashed circles: testis lobule boundary. Nuclear morphologies are characteristic of each cell type[5] and labeled in the image as (1) GSC, (2) spermatocyte, (3) round spermatid, (4) elongating spermatid, and (5) sperm. Stage 2-5 are considered as differentiated germ cells. Note the increased fraction of EdU+ nuclei and reduced number of nuclei corresponding to differentiated germ cells after RNAi compared to controls. Quantifications of EdU+ nuclei abundance, GSC number density, and the fraction of differentiated germ cells are provided in Fig. 3c-e along with other RNAi experiments for direct comparison. **d, e** FISH images showing broader expression of *nanos-1* (**d**) and *eled* (**e**) in testes after *oc-1* RNAi. Arrowheads: differentiated germ cells that do not express *nanos-1* or *eled*. *n*: number of samples exhibiting the reported phenotype out of the total number of samples analyzed. RNAi experiments were repeated on at least three biological replicates.

**oc-1 homeobox transcription factor suppresses proliferation and promotes differentiation of male GSCs.** To understand the function of the genes specifically expressed in *nanos-1*+ cells, we first studied the only three TFs that were identified in this population: *oc-1* in both GSCs and S1 cells, *nr* only in S1 cells, and a homolog of iroquois homeobox TF (*irx*, Smp_063520) in GSCs. Whole-mount in situ hybridization (WISH) confirmed their expression (Fig. 2a). *oc-1* is expressed in both primordial testes and vitellaria, *nr* is specific to primordial vitellaria, and *irx* is expressed in male and female gonadal primordia. Fluorescence in situ hybridization (FISH) experiments further showed that *oc-1* is expressed in *nanos-1*+ GSCs in testes (Supplementary Fig. 3a).

We then performed RNAi to knock them down in juvenile parasites. For these experiments, parasites were retrieved from mice 3.5 weeks post-infection and soaked in double-stranded RNA (dsRNA) continuously in vitro for 2 weeks. We focused on the male germline because female development is retarded under in vitro culture and therefore associated phenotypes are difficult to evaluate[5].

While neither *nr* nor *irx* RNAi showed discernible phenotypes (Supplementary Fig. 4), *oc-1* RNAi led to a significant increase in the number of GSCs and a marked reduction in the abundance of differentiated germ cells, including spermatocytes, spermatids, and sperm (Fig. 2b,c). Consistently, GSC markers including *nanos-1* and *eled* were expressed more broadly in testes after *oc-1* RNAi (Fig. 2d, e). The expansion of the GSC population caused these cells to be packed at higher density compared to control samples, though the size of the testis lobules remained mostly unchanged. The majority of GSC nuclei were also labeled by 5-Ethynyl-2′-deoxyuridine (EdU) after *oc-1* RNAi, suggesting that they were active in DNA synthesis. To test whether the loss of differentiated germ cells was due to the lack of differentiation or apoptosis, we performed TUNEL staining and noticed no increase in apoptotic activity after *oc-1* RNAi beyond the baseline level in control animals (Supplementary Fig. 5). Together, these results suggest that *oc-1* is required for preventing GSCs from over-proliferating and for promoting the initiation of the differentiation program.

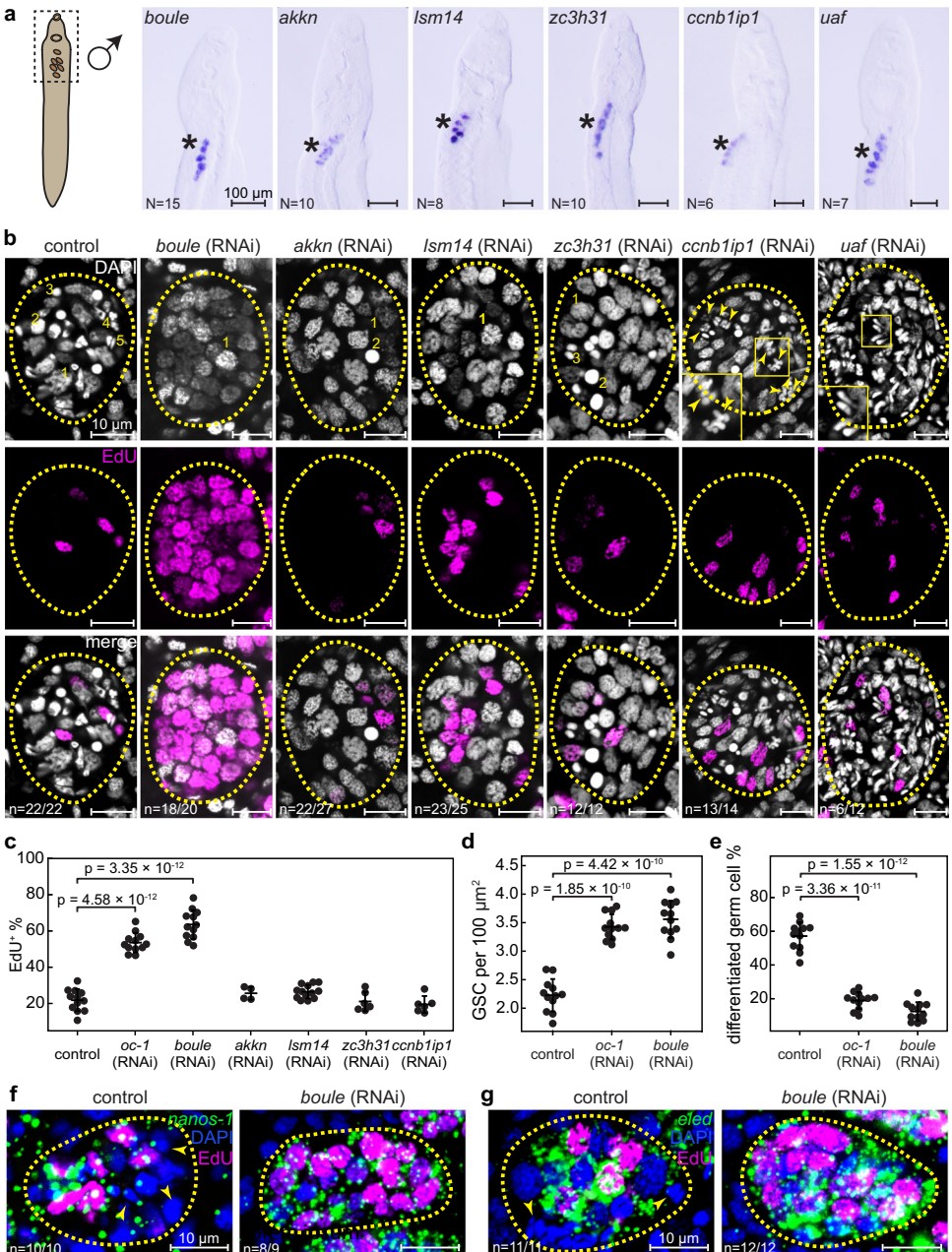

**Fig. 3 An RNAi screen identifies that *boule* phenocopies *oc-1* after RNAi in the juvenile male germline. a** WISH images showing the expression of *boule*, *akkn*, *lsm14*, *zc3h31*, *ccnb1ip1*, and *uaf* in male juvenile parasites. Asterisks: testes. The imaged areas correspond to the dashed box in the schematic on the left. *N*: number of samples showing similar results over two independent experiments. **b** Confocal images showing representative individual testis lobules stained by DAPI and EdU in control and *boule*, *akkn*, *lsm14*, *zc3h31*, *ccnb1ip1*, and *uaf* RNAi juvenile parasites. Insets: magnified boxed areas. Dashed circles: testis lobule boundary. Arrowheads in *ccnb1ip1* RNAi image: incomplete separation of meiotic nuclei. Nuclear morphologies are labeled in the images as in Fig. 2c. **c** Fraction of EdU+ nuclei, **d** number density of GSCs and **e** fraction of differentiated germ cells in testes in control and RNAi parasites. Each data point represents the mean across all testis lobules in a single parasite. *N* = 12 (control RNAi); *N* = 12 (*oc-1* RNAi), *N* = 12 (*boule* RNAi), *N* = 4 (*akkn* RNAi); *N* = 13 (*lsm1* RNAi); *N* = 6 (*zc3h31* RNAi); *N* = 6 (*ccnb1ip1* RNAi). Data are plotted as average ± standard deviation and p-values are calculated using two-sided Welch's *t*-test. **f**, **g** FISH images showing broader expression of *nanos-1* (**f**) and *eled* (**g**) in testes after *boule* RNAi. Arrowheads: differentiated germ cells that do not express *nanos-1* or *eled*. *n*: number of samples exhibiting the reported phenotype out of the total number of samples analyzed. RNAi experiments were repeated on at least three biological replicates.

**An RNAi screen identifies boule as a germline regulator that phenocopies *oc-1* after knockdown.** To identify the pathway through which oc-1 functions, we performed an RNAi screen to knockdown genes enriched in GSCs identified by scRNAseq, focusing on predicted RNA binding proteins, enzymes, and receptors. The goal of these experiments was to identify genes that phenocopy *oc-1* upon RNAi. Out of the 39 genes knocked down (Supplementary Data 2), we observed male germline developmental defects in RNAi experiments of 6 genes (Fig. 3a). Of these, only the knockdown of *boule* gave rise to a similar but stronger phenotype compared to that of *oc-1* RNAi (Fig. 3b–g): testis lobules contained EdU+ *nanos-1*+/*eled*+ GSCs at a higher density compared to controls and drastically reduced number of differentiated germ cells. Consistent with its phenotype, we

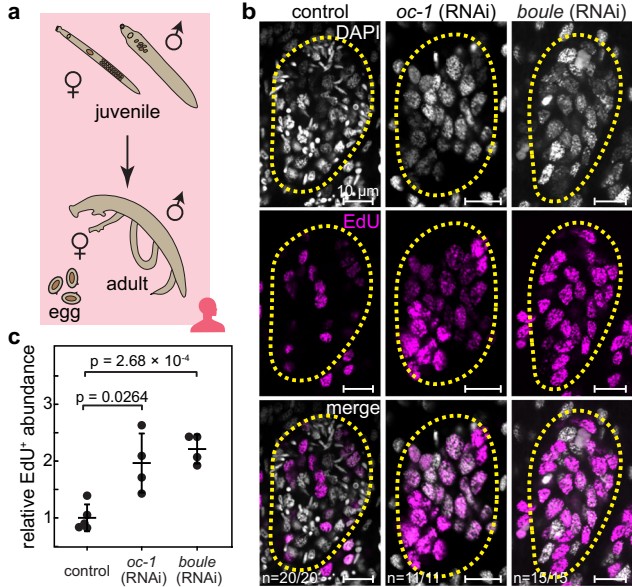

**Fig. 4 Knockdowns of *boule* and *oc-1* cause male GSC over-proliferation in adult parasites. a** Schematic of the sexual maturation process from juveniles to adults, which mate and lay eggs. **b** Confocal images showing representative individual testis lobules stained by DAPI and EdU in adult parasites after control, *oc-1*, and *boule* RNAi. Dashed circles: testis lobule boundary. Note a significant increase of EdU⁺ nuclei and a marked reduction of differentiated germ cells. *n*: number of samples exhibiting the reported phenotype out of the total number of samples analyzed. **c** Since accurate counting of germ cells, which are densely packed in adult testes, is challenging, the number density of EdU⁺ nuclei per unit image area in testes normalized against the abundance in control parasites is quantified instead. $N = 5$ (control RNAi); $N = 4$ (*oc-1* RNAi); $N = 4$ (*boule* RNAi). Data are plotted as average ± standard deviation and *p*-values are calculated using two-sided Welch's *t*-test. RNAi experiments were repeated on at least three biological replicates.

noticed that *boule* is co-expressed with *oc-1* in the male GSCs (Supplementary Fig. 3). boule is a member of the Deleted in Azoospermia (DAZ) family of RNA-binding proteins, which are known germline regulators with important roles in both GSC maintenance and gametogenesis[26–31]. To investigate whether other genes associated with DAZ function in other animals are also involved in schistosome germline development, we searched for potential DAZ-interacting partners, DAZAP and DZIP[28,49,50], in the *S. mansoni* genome based on BLAST similarity. We only found one DZIP homolog, RNAi of which did not generate a germline phenotype (Supplementary Fig. 6), indicating that boule function in the schistosome is either independent of DZIP or involves other unidentified redundant interacting proteins.

The other five genes identified in this screen affected germline differentiation but not proliferation. RNAi of a putative ankyrin repeat-containing protein kinase (*akkn*, Smp_131630) and an LSM14 domain-containing protein (*lsm14*, Smp_129960) blocked the differentiation of germ cells. RNAi of a zinc finger CCCH domain-containing protein (*zc3h31*, Smp_056280) and an E3 ubiquitin-protein ligase CCNB1IP1 (*ccnb1ip1*, Smp_344490) caused defects in spermatogenesis. RNAi of a homolog of uro adherence factor A (*uaf*, Smp_105780) resulted in premature accumulation of sperm and abnormal GSC nuclear morphology (Fig. 3b, c).

Finally, to examine whether these genes play similar functional roles in homeostasis beyond development, we performed RNAi experiments on sexually mature parasites, retrieved from mice 5.5 weeks post-infection (Fig. 4a). Figure 4b, c shows that most

phenotypes, including those induced by *oc-1* and *boule* knock-downs, were identical between juveniles and adults, except for *akkn*, which did not yield a loss-of-function phenotype in adult parasites (Supplementary Fig. 7), suggesting that its function is restricted to regulating development.

***oc-1* and *boule* expressions are mutually dependent and exhibit similar genetic interactions with *nanos* and *eled*.** The observation that RNAi of *oc-1* and *boule* led to similar phenotypes motivated us to investigate the nature of their genetic relation. We followed up RNAi of either gene with WISH to evaluate the changes in the expression of both. We found that *boule* expression was undetectable after *oc-1* RNAi, and *oc-1* expression was also eliminated upon *boule* RNAi (Fig. 5a). This result suggests that *oc-1* and *boule* regulate each other (directly or indirectly) at the transcriptional level.

If *oc-1* and *boule* overlap through the same functional pathway, we expect them to exhibit identical epistatic interactions with other GSC regulators. Previously, we found that *nanos-1* is required for GSC proliferation and differentiation, knockdown of which causes degeneration of testes and loss of differentiated germ cells[5]. In contrast, *eled* is thought to inhibit GSC differentiation, RNAi of which ectopically activates spermatogenesis and leads to premature accumulation of sperm[5]. As their RNAi phenotypes are distinct from those resulting from *oc-1* and *boule* RNAi, it is feasible to test genetic interactions among this set of genes through double RNAi knockdowns. In order to rule out the possibility that treating worms with multiple dsRNA sequences might reduce the knockdown efficacy due to their competition for the RNAi machinery, we also validated the efficient gene knockdown in double RNAi experiments (Supplementary Fig. 8).

Using this approach, we found that both *nanos-1;oc-1* and *nanos-1;boule* double knockdowns exhibited the phenotype of *nanos-1* single knockdown (Fig. 5b), suggesting that *nanos-1* likely functions upstream of *oc-1* and *boule* as a genetic suppressor. By contrast, *eled;oc-1* or *eled;boule* double knock-downs alleviated the defects observed in *oc-1* or *boule* single knockdowns. In particular, the number of EdU⁺ GSC nuclei was significantly reduced after knocking down *eled* along with *oc-1* or *boule* (Fig. 5b, c). These observations indicate that *oc-1*, *boule* and *eled* collectively maintain the balance between differentiation and proliferation of GSCs, with *oc-1* and *boule* suppressing prolifera-tion and promoting differentiation versus *eled* suppressing differentiation (Fig. 5d). When the functions of both sides were lost after double knockdowns, the balance was reestablished.

**Planarian homolog of *onecut* is also required for male germline development.** To determine whether the function of *onecut* in germline development is evolutionarily conserved, we studied the planarian flatworm *S. mediterranea*, a free-living cousin of schistosomes. Using BLAST sequence similarity search, we identified three planarian *onecut* homologs (Supplementary Fig. 2), but only one (*Smed-oc-1*, dd_Smes_v1_39638_1_1) was detected in the male germline (Supplementary Fig. 9a). Specifi-cally, it is expressed in testis lobules, which are distributed beneath the dorsal epidermis of sexually mature planarians; in sexually immature planarians, *Smed-oc-1* expression is specific to testis primordia, which are essentially GSC clusters (Fig. 6a, b). High-resolution images of individual testis lobules showed that *Smed-oc-1* expression is restricted to GSCs (spermatogonial stem cells) and mitotic spermatogonia, but not detected in differ-entiated germ cells (Fig. 6c). Consistently, in asexual planarians, *Smed-oc-1* is also present in presumptive GSCs (Supplementary Fig. 9b).

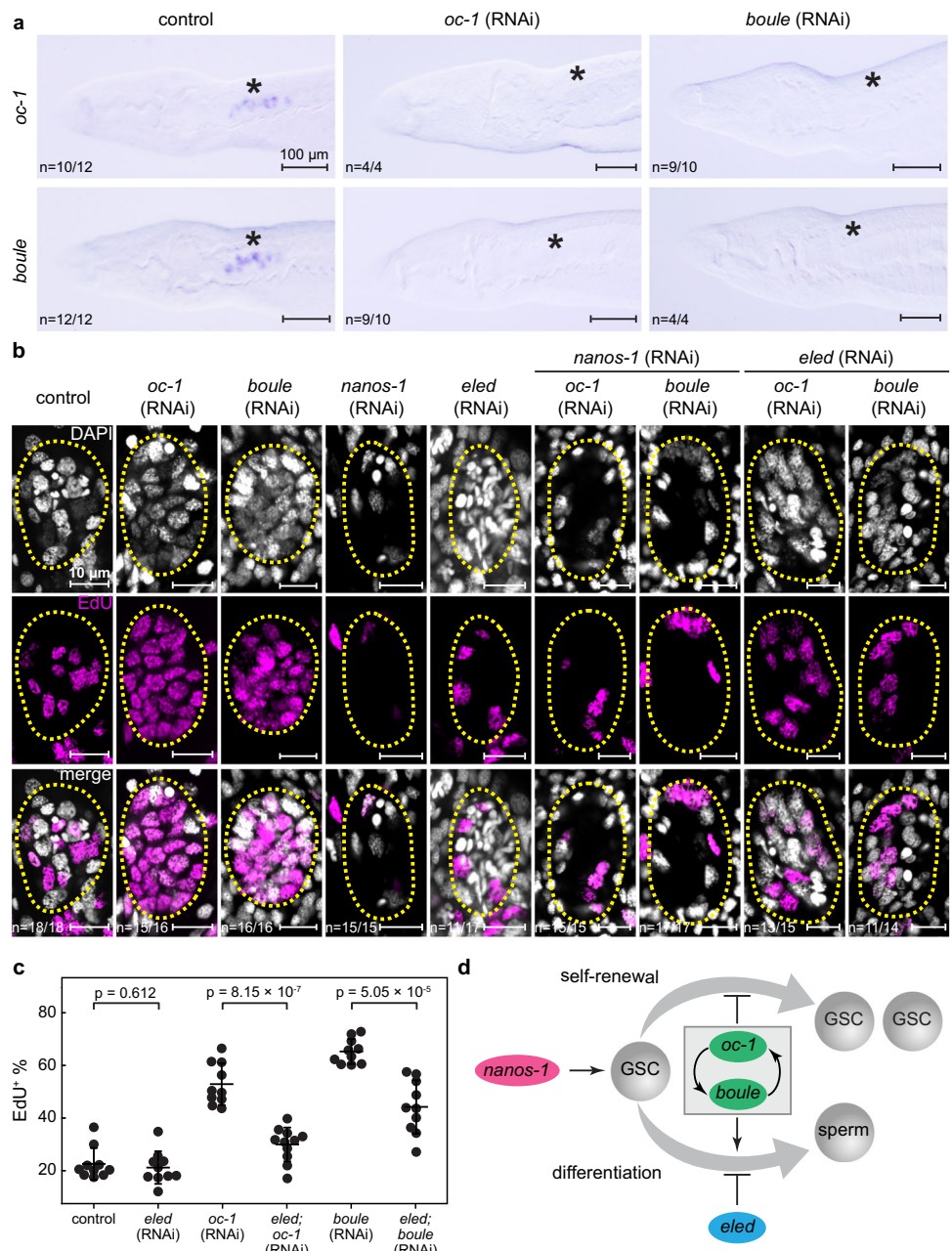

**Fig. 5 Epistatic interactions of *oc-1*, *boule*, *nanos-1*, and *eled*. a** WISH images showing that the expression of *oc-1* and *boule* are mutually dependent. Knockdown of either one causes the elimination of both. Asterisks: testes. **b** Confocal images showing representative individual testis lobules stained by DAPI and EdU in juvenile parasites after RNAi treatments. Dashed circles: testis lobule boundary. *n*: number of samples exhibiting the reported phenotype out of the total number of samples analyzed. **c** Fraction of EdU+ nuclei in testes in control and RNAi parasites. Each data point represents the average across all testis lobules in a single parasite. $N = 10$ (control RNAi); $N = 10$ (*eled* RNAi); $N = 10$ (*oc-1* RNAi); $N = 11$ (*eled;oc-1* RNAi); $N = 10$ (*boule* RNAi); $N = 10$ (*eled;boule* RNAi). *nanos-1* RNAi experiments were not quantified because only a few GSC nuclei are present per testis lobule, which is insufficient for reliable statistics. Data are plotted as average ± standard deviation and *p*-values are calculated using two-sided Welch's *t*-test. RNAi experiments were repeated on at least three biological replicates. **d** Proposed model of genetic interactions between *oc-1*, *boule*, *nanos-1*, and *eled* in regulating male GSC identity, proliferation, and differentiation.

Knockdown of *Smed-oc-1* led to the complete loss of testes (Fig. 6d). Even in early stages of RNAi, the erosion of the spermatogonial layer had already began, while differentiated germ cells were largely unaffected (Fig. 6e), suggesting that *Smed-oc-1* is required for the maintenance of GSCs and spermatogonia. This function is distinct from that of *Sm-oc-1*, which is dispensable for maintaining the identity of GSCs but necessary for regulating GSC proliferation and differentiation.

We next evaluated the genetic relations between *Smed-oc-1* and other known planarian germline regulators such as *nanos* and *boule*. The planarian *nanos* homolog, *Smed-nanos*, is a GSC marker and required for GSC specification and maintenance[17,24]. The planarian *boule* homologs, *Smed-boule-1* and *Smed-boule-2*, are expressed in GSCs and spermatogonia. *Smed-boule-1* is involved in spermatogenesis and loss of its function leads to the elimination of differentiated germ cells, whereas *Smed-boule-2* is necessary for the maintenance of GSCs and spermatogonia

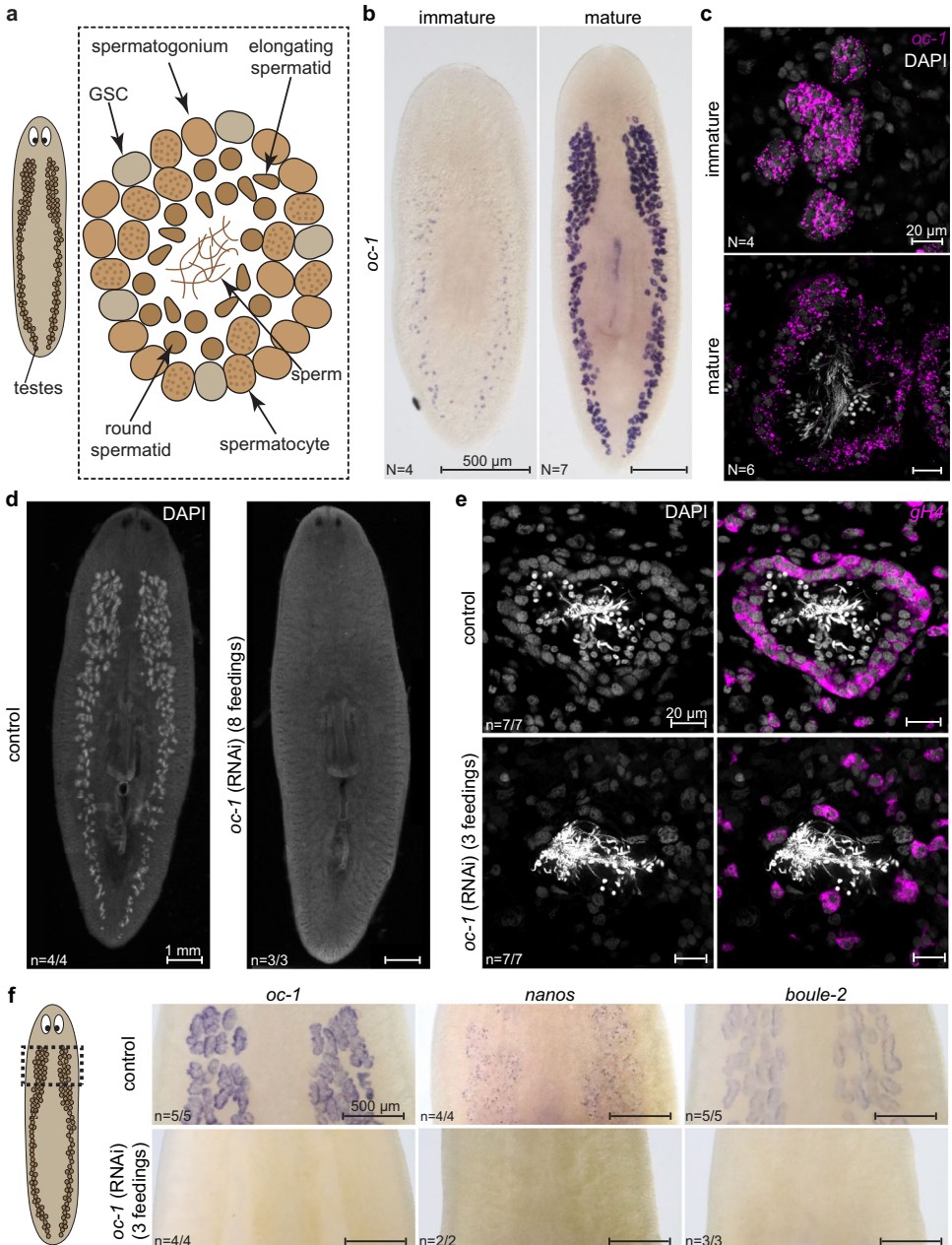

**Fig. 6 *Smed-oc-1* is required for the maintenance of spermatogonial cells. a** Schematic of planarian testes lobules containing spermatogonial cells (GSCs and mitotic spermatogonia) at the periphery and differentiated germ cells (spermatocytes, round spermatids, elongating spermatids, and sperm) distributed in the internal layers. **b** WISH images showing the expression of *Smed-oc-1* in sexually immature (left) and mature (right) planarians. The punctuated signal in the immature planarian corresponds to presumptive testis primordia. In the mature planarian, the signal is specific to testes. **c** FISH images of *Smed-oc-1* showing its expression in testis primordia within sexually immature planarians (top) and in spermatogonial cells (including GSCs and mitotic spermatogonia) in sexually mature planarians (bottom). *N*: number of samples showing similar results over three independent experiments.
**d** Dorsal view of sexually mature planarians after control and *Smed-oc-1* RNAi. Note that testes, which are distributed beneath the dorsal epithelium and visualized by strong DAPI signal, are completely eliminated after *Smed-oc-1* RNAi. Animals were fixed after eight feedings of dsRNA. **e** Testes of control (left) and *Smed-oc-1* (right) RNAi planarians after three feedings of dsRNA. In control animals, *Smed-germinal histone H4* (*gH4*) labels the spermatogonial cells at the periphery of testis lobules, which are mostly absent after *Smed-oc-1* RNAi, while differentiated germ cells remain at the center of the testis lobules. *gH4* expression in individually isolated somatic stem cells maintains after *Smed-oc-1* RNAi. **f** WISH images showing that the expression of *Smed-oc-1*, *Smed-nanos*, and *Smed-boule-2* are disrupted after three feedings of *Smed-oc-1* RNAi. The imaged areas correspond to the dashed box in the schematic on the left. *n*: number of samples exhibiting the reported phenotype out of the total number of samples analyzed. RNAi experiments were repeated on at least three biological replicates, each containing a group of planarians fed with dsRNA separately.

(Supplementary Fig. 9c)[28]. We found that, *Smed-oc-1* RNAi quickly abolished *Smed-nanos* and *Smed-boule-2* expression (Fig. 6f). These results suggest that *Smed-oc-1* function overlaps with that of *Smed-nanos* and *Smed-boule-2*, but not *Smed-boule-1*. Based on these observations, we concluded that the expression of *onecut*, *boule,* and *nanos* in the male germline appears to be conserved between planarian and schistosome, but the functional relationships within this gene set have been rewired significantly through evolution.

## Discussion

Schistosome germline is essential for both disease transmission and pathology, as it produces eggs that serve as the transmissible agents and induce host pathological inflammatory response[13]. The germline develops post-embryonically at the juvenile stage from a population of stem cells with a somatic origin that is carried over from asexual larval parasites[5]. During the germline development, GSCs need to be regulated to perform their two essential functions: proliferation, which produces more daughter stem cells through mitotic divisions (self-renewal), and differentiation, which initiates meiosis towards the production of gametes. In this work, we identified a germline-specific stem cell regulatory program that balances the fate of GSCs between proliferation and differentiation. We also studied the evolutionary conservation and modification of this program by comparing the schistosome and its planarian cousin.

Our study used scRNAseq to construct a complete map of the juvenile schistosome stem cell population. The stem cells were separated into nine populations, which were further divided into two groups based on their transcriptional signatures: the presumptive multipotent populations, including the GSCs, and the presumptive tissue-specific progenitors of muscle, neural, intestinal/parenchymal, and epidermal lineages. Each progenitor population expresses a specific battery of genes associated with respective differentiated cell types, whereas multipotent populations exhibited global repression of these differentiation genes. This stem cell subpopulation structure is strikingly similar to that of planarian stem cells (which also contain pluripotent and lineage-restricted progenitors[51–53]), raising the question of whether these subpopulations have a one-to-one correspondence between the two animals. In this regard, our work establishes a model system and provides the data needed for comparative studies aiming at understanding stem cell evolution at the single-cell transcriptomic level.

The transcriptomic characterization of the schistosome GSCs led us to the discovery of a conserved regulatory gene set in the male germline, with onecut homeobox transcription factor and boule mRNA binding protein at its core. In the schistosome, *onecut* and *boule* expressions are mutually dependent; they form a genetic circuit that inhibits the over-proliferation of GSCs and promotes differentiation, balancing the effect of *eled*, which suppresses differentiation of GSCs. In the planarian, homologs of *onecut* and *boule* (specifically *Smed-oc-1* and *Smed-boule-2*) are instead required for GSC maintenance. Compared to the planarian, which has a low fertility rate, the schistosome has impressive fecundity with each pair of sexually mature parasites being capable of laying hundreds of fertilized eggs per day[54,55]. Whether the regulatory modifications around *onecut* and *boule* are associated with the difference in reproductive output is an important question for future research. Our study lays the foundation by identifying the key members of this previously uncharacterized gene circuit that regulates GSC fate.

We expect the conservation of this germline-specific regulatory program to extend beyond flatworms. boule homologs, along with other members of the DAZ protein family, are known germline regulators in diverse animals from basal invertebrates to humans[26–31], though their premeiotic functions in GSCs as reported here are considered to be rare among invertebrates[28]. By contrast, onecut homologs have been studied in the context of neural, liver, and pancreatic development using different animal models[33–38], but its function in germline development and its genetic interaction with *boule* were previously unknown. Of interest to our study is the observation that onecut homologs are indeed expressed in rodent and human testes[32–35], suggesting that its germline function may be more universal than previously acknowledged.

An important question raised by our results is what factors maintain the identity of GSCs. Our previous work suggests that GSCs are specified from *eled*[+] stem cells during the early juvenile development by activating *nanos-1* expression[5]. In the current scRNAseq data set, we noticed that GSCs are distinguished from other stem cell populations in the presumptive multipotent group only by a small number of genes, while sharing a large set of common transcriptional signatures. The GSC-specific genes only contain two TFs (*oc-1* and *irx*), but neither of them is required for maintaining GSC identity. Knockdown of other genes enriched in GSCs mostly caused defects in differentiation and spermatogenesis but did not affect the identity of GSCs either. These observations favor the hypothesis that GSC identity may be maintained by extrinsic cues, though regulation by other intrinsic determinants such as post-transcriptional regulatory factors cannot be excluded.

In mammalian testes, these extrinsic cues are provided by discrete somatic niche cells (e.g., Sertoli cells) to support male GSCs[56,57]; in *Drosophila* and *C. elegans*, stromal cap and distal tip cells form the niche, respectively, instructing GSC fates through several signaling pathways[58,59]; in the planarian, somatic gonadal cells required for male GSC specification and maintenance have also been defined[60,61]. All these "niche" cells are in close spatial proximity with their GSC counterparts. However, in the schistosome testes, most GSCs are not in contact with any somatic tissues. This is particularly clear in *oc-1* RNAi or *boule* RNAi animals where many testis lobules contain exclusively a large number of densely packed GSCs. These observations suggest that the extrinsic cues for maintaining GSC identity in schistosomes are likely systemic, potentially through neural or hormonal regulations. This hypothesis is particularly compelling because schistosome maturation is known to depend on host immune cues[12,13]. The host signals need to be sensed by parasite cells at the host-parasite interface, then translated and relayed internally to instruct GSC activities[62]. In the planarian, germline development is regulated by neuroendocrine signals, such as neuropeptides[63], implicating that there may be similar crosstalk between the nervous system and the gonads in schistosomes. Here our characterization of the intrinsic regulatory program of the schistosome GSCs opens the door for exploring these important questions. Given the significance of parasite germline in disease transmission and pathology, a better understanding of the basic biology of this developmental process will help to define vulnerable points for combating this major parasitic disease.

## Methods

**Animals**. *S. mansoni* juveniles and adults were retrieved from infected female Swiss Webster mice (NR-21963) by hepatic portal vein perfusion with 37 °C DMEM supplemented with 5% heat-inactivated FBS at 3.5 weeks and 5.5 weeks post-infection, respectively. Worms were cultured at 37 °C in Basch Media 169 supplemented with 1× Antibiotic-Antimycotic[11]. In adherence to the Animal Welfare Act and the Public Health Service Policy on Humane Care and Use of Laboratory Animals, all experiments with and care of mice were performed in accordance with protocols approved by the Institutional Animal Care and Use Committees (IACUC) of Stanford University (protocol approval number 30366).

Sexual and asexual *S. mediterranea* were maintained in the dark at 18 °C in 0.75× Montjuïc salts or 0.5 g/L Instant Ocean Sea Salts supplemented with 0.1 g/L sodium bicarbonate, respectively. Planarians were fed calf liver paste once or twice weekly and starved at least 5 d prior to fixation. For RNAi experiments, large sexual planarians were chosen to ensure sexual maturity. Smaller sexual animals (<4 mm in length) were used to test expression in sexually immature animals.

**scRNAseq and data analysis**. *S. mansoni* juveniles were transferred into 15 mL conical tubes and rinsed twice with PBS pre-warmed at 37 °C and dissociated into single-cell suspensions in 4 mL of 0.25% trypsin in HBSS for 20 min. Cell suspensions were passed through a 100 µm nylon mesh (Falcon Cell Strainer) and immediately doused with 11 mL of PBS supplemented with 1% BSA. The sample was then centrifuged at 300×g for 5 min at 4 °C. Pellets were gently resuspended in PBS with 0.5% BSA, passed through a 30 µm nylon mesh, and stained with 5 µg/mL DAPI for 30 min. Stained samples were then washed and resuspended in PBS with 0.5% BSA and loaded on a SONY SH800S cell sorter. Dead cells were excluded based on DAPI fluorescence. Droplets containing single cells were gated using forward scattering (FSC) and side scattering (SSC). Cells that passed these gates were sorted into 384-well lysis plates containing Triton X-100, ERCC standards, oligo-dT, dNTP, and RNase inhibitor.

Reverse transcription and cDNA pre-amplification were processed with the Smart-seq2 protocol[25]. We performed 23 cycles of PCR amplification and diluted the resulting cDNA in Qiagen EB buffer at 1:20 dilution as we empirically determined to yield an average final concentration of 0.4 ng/µL. Diluted cDNA was then tagmented and barcoded using in-house Tn5 tagmentase and custom barcodes fitted for 384-well plate. Library fragments concentration and purity were quantified by Agilent bioanalyzer and qPCR. 2 × 150 bp paired-end sequencing was performed on a NovaSeq 6000 at a depth of ~500,000 reads per cell at the Chan Zuckerberg Biohub Genomics core.

Raw sequencing reads were demultiplexed and converted to fastq files using bcl2fastq. Paired-end reads were mapped to *S. mansoni* reference transcriptome (WormBase Parasite) using Salmon (version 0.14.0) with "-validateMappings" flag. We performed downstream preprocessing and analyses on the estimated read counts in the Salmon output. To eliminate low-quality cells, we filtered out cells with fewer than 519 genes detected, passing 7657 cells for downstream analysis. This cut-off was chosen to bisect the bimodal distribution of the number of genes detected per cell.

Raw gene counts were normalized for sequencing coverage such that each cell has a total number of counts equal to the median library size of all cells. The resulting data were added with a pseudocount of 1 and Log2-transformed. Processed gene expressions with values less than 1 were set to zero, and genes detected in <1% or >99% of cells were filtered out. The SAM algorithm (version 0.7.5) was run with parameters 'weight_PCs=False' and 'preprocessing=Normalizer'[25]. SAM outputs gene weights, principal component (PC) coordinates, a nearest-neighbor graph, and 2D UMAP projections used for visualization, on which the processed gene expression data can be also overlaid.

Sub-clustering the stem cell populations was done by running SAM on *ago2-1*[+] cells. We used the Leiden algorithm (version 0.7.0)[64] to determine the clusters and annotated their respective identity manually based on marker gene expression. We computed the centroid of each cluster in the PC space output by SAM. The PC space was computed using the top 3000 genes for the full data set, and the top 8000 genes for the stem cell subpopulations. These numbers were chosen by SAM based on the size of the data sets to satisfy computational memory constraints of the algorithm[25]. We performed hierarchical clustering through the average linkage method using the correlation distances between the PC centroids to generate the dendrogram shown in Fig. 1c.

For each cluster, marker genes were identified by computing partial sums of gene dispersions from the *k*-nearest-neighbor-averaged data:

$$S_{ix} = \sum_{j=1}^{n_x} \left( E_{ji} - \mu_i \right)^2 / \mu_i \qquad (1)$$

where $E_{ji}$ is the nearest neighbor averaged expression value for cell *j* and gene *i*, $\mu_i$ is the mean expression of gene *i*, $n_x$ is the set of cells in cluster *x*, and $S_{ix}$ is the unnormalized maker score for gene *i* in cluster *x*. To account for differences across clusters, we normalized the marker score by the maximum value, $Z_{ix} = S_{ix} / \max_i\{S_{ix}\}$. We ranked the genes based on their marker scores, and only kept those with >0.5 SAM weight, >0.1 normalized marker score, and >2 fold enrichment with respect to cells outside the cluster. The complete list of marker genes is provided in Supplementary Data 1. For ease of visualization, only the top 50 markers for each stem cell cluster are shown in Fig. 1c.

**RNAi**. To synthesize dsRNA, gene fragments were amplified from cDNA using oligonucleotide primers listed in Supplementary Data 2, 3 and cloned into the vector pJC53.2 (Addgene Plasmid ID: 26536)[60]. 10–15 juvenile or 6–8 adult parasites were soaked in Basch media 169 supplemented with ~20 µg/mL dsRNA against each target for 2 weeks. The media with dsRNA were refreshed daily. Each RNAi was performed with at least two biological replicates each containing at least two technical replicates to evaluate phenotypical changes. A small population of parasites exhibiting visible damage or reduced mobility via microscopic

examination was excluded from downstream analyses. Potential off-target effects were ruled out by using dsRNA sequences to target non-overlapping regimes of the same gene of interest (Supplementary Fig. 10). For gene knockdown in planarians, dsRNA was mixed with liver paste at a concentration of 70 ng/µL dsRNA and fed to planarians every 4 days for 3–8 feedings. In all RNAi experiments, dsRNA matching the *ccdB* and *camR*-containing insert of pJC53.2 was used as the negative control. The two-sided Welch's *t*-test and *p*-value calculation were performed via *t*-test in sciPy library with default parameters.

**In situ hybridization**. RNA WISH and FISH experiments were performed following the established protocol[5,25,65]. Briefly, schistosomes were killed in 6 M MgCl₂ for 30 s, fixed in 4% formaldehyde supplemented with 0.2% Triton X-100 and 1% NP-40 for 4 h, and then dehydrated in methanol. Adult parasites were relaxed by 0.25% ethyl 3-aminobenzoate methanesulfonate for ~1 min to separate male and female parasites before fixation. Dehydrated juveniles were bleached in 3% H₂O₂ in methanol for 30 min, rehydrated, permeabilized by 10 µg/mL proteinase K for 20 min, and then post fixed with 4% formaldehyde. Dehydrated adults were rehydrated, bleached in the bleaching solution (5% formamide, 0.5× SSC, 1.2% H₂O₂) for 45 min under bright light, permeabilized by 10 µg/mL proteinase K for 30 min, and then post fixed with 4% formaldehyde. The hybridization was performed at 52 °C with riboprobes, which were then detected through either alkaline phosphatase-catalyzed NBT/BCIP reaction (for WISH) or peroxidase-based tyramide signal amplification (for FISH).

Planarians were relaxed on ice and killed in 5% *N*-Acetyl Cysteine (NAC) solution for 10 min. They were fixed for 2 h in 4% formaldehyde supplemented with 1% NP-40, dehydrated in methanol, and stored at −20 °C. Samples were then rehydrated, bleached for 2 h in the bleaching solution under bright light, permeabilized with 10 µg/mL proteinase K for 10 min, and post fixed with 4% formaldehyde. The rest of the steps were performed similarly to those described for schistosome in situ hybridization experiments, except that the hybridization was performed at 56 °C. To compare gene expression between control and RNAi conditions using WISH, the development of WISH signal was progressed in parallel and simultaneously stopped.

**EdU labeling**. EdU (TCI Chemicals) was dissolved in DMSO at 10 mM and added into the medium at a final concentration of 10 µM. Both juvenile and adult schistosomes were pulsed with EdU overnight. Parasites were fixed and permeabilized as in in situ hybridization experiments. EdU incorporation was detected by click reaction with 25 µM of Cy5-azide conjugates or Carboxyrhodamine 110 Azide conjugates (Click Chemistry Tools)[25]. To combine EdU detection with FISH, click reaction was performed after tyramide signal amplification.

**TUNEL assay**. Juvenile parasites were fixed and permeabilized as in in situ hybridization experiments. We used the ApoTag Fluorescence In Situ Detection Kit (Millipore Sigma) for TUNEL staining. Six juvenile worms were incubated in 30 µL of 30:70 (v/v) TdT Enzyme: Reaction Buffer at 37 °C for 4 h. The reaction was then stopped by adding 1:34 (v/v) Stop/Wash Buffer: H₂O for 5 min at room temperature. After washes, samples were treated in 5% horse serum and 0.5% Roche Western Blocking Reagent for 1 h, and then incubated in 53:47 (v/v) Blocking Solution: Anti-Digoxigenin Conjugate at room temperature for 4 h.

**Imaging**. Samples for fluorescence imaging were mounted in the scale solution (30% glycerol, 0.1% Triton X-100, 2 mg/mL sodium ascorbate, 4 M urea in PBS)[66] and imaged on a Zeiss LSM 800 confocal microscope using either a ×20 (N.A. = 1.0, working distance = 1.8 mm) water-immersion objective (W Plan-Apochromat) or a ×40 (N.A. = 1.1, working distance = 0.62 mm) water-immersion objective (LD C-Apochromat Corr M27). To obtain confocal images of schistosome testes, the midplane of testis lobules was first determined, then ~7 confocal sections with optimal z spacing recommended by Zen software were recorded around the midplane to generate maximum intensity projections. WISH samples were mounted in 80% glycerol with 10 mM Tris supplemented with 1 mM EDTA, pH = 7.5 for imaging.

**Reporting summary**. Further information on research design is available in the Nature Research Reporting Summary linked to this article.

## Data availability

The authors declare that all data supporting the findings of this study are available within the article and its supplementary information files or from the corresponding author upon reasonable request. The schistosome stem cell scRNAseq data generated in this study have been deposited in the Gene Expression Omnibus (GEO) database under accession code: GSE147355.

## Code availability

The SAM analysis source code, tutorials, and documentation are available through Github [https://github.com/atarashansky/self-assembling-manifold].

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

## Acknowledgements

*S. mansoni* (strain: NMRI) was provided by the NIAID Schistosomiasis Resource Center for distribution through BEI Resources, NIH-NIAID Contract HHSN272201000005I. We thank M. Khariton, Y. Fan, S. Korullu, and R. Jones for experimental help, and J. Lee and J. Gao for the critical comments on the manuscript. P.L. is partially supported by an Agilent Bioengineering Fellowship. D.N.S, Y.X, and A.J.T. are supported by the Stanford Bio-X Graduate Fellowships. B.W. is supported by a Beckman Young Investigator Award.

## Author contributions

P.L. and B.W. designed the research, Y.X. and P.L. performed the cell sorting and scRNAseq experiments, Y.X., A.J.T., and P.L. analyzed the scRNAseq data, P.L., D.N.S., and X.Y. performed the functional experiments, P.L. and B.W. wrote the paper with input from all other authors, S.R.Q. and B.W. supervised the project and provided conceptual advice.

## Competing interests

The authors declare no competing interests.
