## [Peer Review File · Nature Communications]

Reviewers' Comments:

Reviewer #1:

Remarks to the Author:

In this work, Li and colleagues use single-cell transcriptomics to identify stem cell and progenitor populations in juvenile schistosomes. Among these populations, they identified a population of Germline Stem Cells (GSCs), which had not been previously characterized at the single-cell level. Functional analyses of marker genes of GSCs uncover a regulatory program that controls their proliferation and differentiation. This regulatory program is controlled by the genes *onecut* and *boule* and is conserved in a close species, *S. mediterranea*, although the function of these genes has diverged.

The work presented by Li and colleagues is interesting and tackles an important point to better understand germline specification and differentiation in *S. mansoni*, which is crucial to develop new treatments against schistosomiasis. However, there are some aspects of the work that need clarification and additional analyses before accepting its publication in its current format.

Major comments

1.- the authors say that they select the stem cells based on the expression of *ago2-1*. However, they do not show the expression of *ago2-1* in the general UMAP plot nor they provide a quantification of *ago2-1* in all the different clusters. I think this would be relevant to the audience to assess the selection of stem cells for further analyses.

2.- the authors applied hierarchical clustering to distinguish their previously identified cell populations (, , and) and GSCs with the newly identified tissue-specific progenitor cells. According to their description in the methods section, they use as a metric the centroid of clusters from the UMAP plot. Although UMAP has been suggested to preserve global data structure, this has been questioned by the scientific community (Dmitry Kobak & George C. Linderman, bioRxiv 2019). Thus, I am not sure if this clustering reflects anything else than the position of cells in the UMAP projection. The authors should justify why they used this approach instead of other conventional methods, such as clustering on the PC or gene space for a set of marker genes for each cluster, or redo the clustering using such a distance metric.

3.- the authors should provide a list of marker genes in all the stem cell and progenitor clusters that they describe in relation to Figure 1c, including also the full list of marker genes identified for the GSCs. For each marker gene, the expression and significance (SAM weight, and fold enrichment) should be provided. Given that the authors only compute these for the stem cell clusters and not for the full dataset, they should also provide these values relative to the full dataset to discard the possibility that these are common genes in juvenile schistosomes.

4.- In Figure 2b, the authors evaluate the effect of *oc-1* RNAi on GSC in testes. Here, the authors note that *oc-1* RNAi increases the amount of EdU+ cells and reduces the amount of differentiated germ cells. The authors should provide a quantification of these changes as they cannot be assessed from a single confocal image.

In page 10, the authors explore the similarity of epistatic interactions between *oc-1* and *boule* with *nanos-1* and *eled*. While it is clear that both genes show similar epistatic interactions with these genes, the nature of these epistatic relation is not clearly shown. The authors should include single *nanos-1* and *eled* RNAi experiments and include the quantification of Edu+ cells in testes in both cases. Additionally, if they want to explore the nature of these epistatic relations (*oc-1/boule* with *nanos-1* and *oc-1/boule* with *eled*), the authors should include data showing similar amount of gene downregulation in both single and double RNAi experiments. Finally, further description of the double RNAi experiments should be included in the methods section.

Minor comments

- 1.- The authors should describe in detail the number of genes that were used to calculate the principal components used to build the UMAP plot.
- 2.- There is a typo in FigS1 legend "FISH image. es: esophagus.
- 3.- The authors should provide quantification of Edu+ cells of all confocal images included in the paper.

Reviewer #2:

Remarks to the Author:

Using scRNAseq, Pengyang et al identify factors expressed in the germline stem cells of *S. Mansion* parasites. Interestingly, there are only few differences between the transcriptomes of these cells versus the neoblast stem cells in the animal, known to give rise to many adult cell types and probably the germ line itself. Using in situ hybridizations to verify enrichment in male or female germline, they describe new factors marking these cell types. RNAi of onecut transcription factor *oc-1* revealed a very interesting phenotype of excess EdU incorporation and reduced differentiation. A similar phenotype was found after inhibition of the RNA-binding factor *boule*, homologs of which had also previously been implicated in Platyhelminth germline maintenance in planarians. *Boule* and *oc-1* were required for each other's expression in the testes, and double RNAi with either *nanos* or *eled* suggested *oc-1* and *boule* act at a similar step in the GSC differentiation pathway, to suppress proliferation and promote differentiation. A homolog of *oc-1* is also expressed in the planarian testes and interestingly is required for germline maintenance. Thus, *oc-1* is a conserved factor that appears to have distinct activities in the two germlines.

Together, the study makes a strongly supported and novel contribution in clarifying the regulatory mechanisms used to control germline maintenance. The work is of high quality and clearly written and makes a nice impact on the field. Because sexual reproduction is a critical aspect of the parasite lifecycle and the germline stem cells in this animal may be maintained through adult stem cells, this would be a novel advance for a wide audience. I have a few issues for authors to consider.

Major comments:

1. Does *oc-1* and *boule* inhibition cause over proliferation or arrested proliferation? It is interesting that these treatments increase EdU staining but do not appear to cause enlargement of that organ. Could *oc-1* and *boule* promote transit through a specific cell cycle stage, such that their inhibition leads to higher EdU uptake but lower output of proliferation and differentiation? A complementary analysis, for example with H3P or FACS, would be helpful to resolve this.
2. The aspect of the *oc-1* and *boule* phenotypes involving loss of differentiation should be more strongly supported either with markers of spermatids or quantifications of the DAPI morphological cell type scoring.
3. The data presented in the double RNAi experiment (Fig 5b and c) is incomplete because there are quantifications of conditions (such as *oc-1* RNAi alone or *boule* RNAi alone) that appear to be missing visualizations in 5b.
4. How were the double RNAi experiments conducted? In other systems, simultaneous treatments with two different dsRNAs can reduce the effects of inhibition from either, leading to the need to have comparative single-gene inhibition conditions dosed with competing control dsRNA in tests of gene interaction. For example, the penetrance or expressivity of animals treated with *oc1*+control dsRNA is likely to be lower than that of *oc1* dsRNA alone, which could impact the interpretations of the study. In that case, low EdU incorporation in the *oc1*+*eled* dsRNA condition could either be due to genetic interaction as authors conclude or due to competition between the *oc1* and *eled* dsRNAs.

There would be a similar issue with the other double RNAi experiments. I'm concerned that without the aforementioned design or some complementary analysis (such as verifying knockdown is equivalent in the single- versus double-RNAi) then it is less certain that the results reveal a genetic interaction.

Minor comments:

1. How does oc-1 and boule expression look like on the tSNE plots as in Fig S1?
2. Only some of the stages are marked in Fig 3b. If some are shown for the knockdown conditions, it would be helpful to present them for the controls as well.
3. Can stages of PGC differentiation be identified through the scRNAseq analysis, for example via trajectory analysis? How do these correspond with states of oc-1 and boule expression?
4. The differences in the role of oc-1 between *S. mediterranea* and *S. mansoni* are intriguing. Could they be due to differences in the knockdown efficiency in these two organisms?

Reviewer #3:

Remarks to the Author:

By using scRNAseq to construct a transcriptomic cell type atlas of juvenile *S. mansoni*, Pengyang Li et al. were succeeded in capturing the GSCs and identifying their transcriptional signatures. By using RNA interference, authors found a genetic program that balances the fate of GSC between proliferation and differentiation which was controlled by onecut and boule.

However, my major concerns about this paper include:

1. Same protocol including cell isolation and Smart-Seq2 program was used for current study and previous studies (ref5, 25). Please highlight how the author made a progress and they were able to detect of lowly expressed genes which were undetectable in previous study (ref25). Are there any parameters indicate those results generated by Smart-Seq2 from lowly expressed genes are reliable data?
2. In this study, RNAi was undertaken by targeting different genes in parasites. To evaluate knockdown efficacy of RNAi, we normally use real time PCR to measure the transcription level of the targeted gene or identify phenotype changes of the targeted protein. Both of those two methods are important to verify the knockdown of targeted gene to ensure the difference observed in the imaging (in situ hybridization or Edu staining) was not induced by off-targeted genes, which is normally presented in dsRNA knockdown system in schistosomes.
3. The same location of oc-1 and nanos-1+ does not mean there are interaction between those two genes. Furthermore the location of nanos-1+ cells in the yolk cell-producing somatic reproductive organ was performed previously (ref 46), not undertaken with the oc-1 in the same experiment, which is hard to compare. Similarly, co-expression of oc-1 and boule in the spermatogonial cells indicated potential interaction presented between those two cells. However, to determine if there is the direct or indirect regulation of oc-1 and boule, more protein functional analysis and protein interaction assays are needed.

A number of points need to be addressed/modified:

1. I suppose that Fig2d (2e) only showed the expression of nanos-1 and elcd after oc-1 RNAi. Please show the control cells without oc-1 RNAi as well. Same issue with Fig 3d and 3e.
2. This stud indicated the roles of oc-1 and boule in suppressing proliferating and promoting differentiation. Can you clarify the relationship between proliferation and differentiation and provide evidences or refs to support it?

Response to referee

We thank the reviewers for their thoughtful and constructive reviews. In response to their comments, we have added new data, revised the text, and modified the figures, as detailed in our responses below. All major changes in the text are highlighted in red. We have tried our best to address all of the referees' comments, and we believe that the manuscript has been improved in the process.

Reviewer #1

Major comments:

1. the authors say that they select the stem cells based on the expression of ago2-1. However, they do not show the expression of ago2-1 in the general UMAP plot nor they provide a quantification of ago2-1 in all the different clusters. I think this would be relevant to the audience to assess the selection of stem cells for further analyses.

Fixed. We now show ago2-1 expression on the UMAP projection in **Fig. 1a**.

2. the authors applied hierarchical clustering to distinguish their previously identified cell populations (μ , δ' , ε_α and ε_β) and GSCs with the newly identified tissue-specific progenitor cells. According to their description in the methods section, they use as a metric the centroid of clusters from the UMAP plot. Although UMAP has been suggested to preserve global data structure, this has been questioned by the scientific community (Dmitry Kobak & George C. Linderman, bioRxiv 2019). Thus, I am not sure if this clustering reflects anything else than the position of cells in the UMAP projection. The authors should justify why they used this approach instead of other conventional methods, such as clustering on the PC or gene space for a set of marker genes for each cluster, or redo the clustering using such a distance metric.

We apologize for the confusion. Our hierarchical clustering was indeed performed in the PC space, as the reviewer suggested. We have revised the methods section to clarify this point (**lines 371-381**).

3. the authors should provide a list of marker genes in all the stem cell and progenitor clusters that they describe in relation to Figure 1c, including also the full list of marker genes identified for the GSCs. For each marker gene, the expression and significance (SAM weight, and fold enrichment) should be provided. Given that the authors only compute these for the stem cell clusters and not for the full dataset, they should also provide these values relative to the full dataset to discard the possibility that these are common genes in juvenile schistosomes.

We have now included all genes that are enriched in subsets of stem cell populations in the revised **Supplementary Table 1**, along with their average expression, SAM weight, enriched population, and fold enrichment. In addition, as described in the text (**lines 102-106**), these genes were identified by comparing across stem cell populations; many in presumptive progenitor clusters are also expressed in corresponding differentiated cell types. To make this clear, we now also specify in the **Supplementary Table 1** whether a gene is specifically expressed in stem cells.

4. In Figure 2b, the authors evaluate the effect of *oc-1* RNAi on GSC in testes. Here, the authors note that *oc-1* RNAi increases the amount of EdU⁺ cells and reduces the amount of differentiated germ cells. The authors should provide a quantification of these changes as they cannot be assessed from a single confocal image.

We now report the percentage of differentiated male germ cells in testes in the new **Fig. 3e**. There is a significant reduction in the number of differentiated germ cells in worms after *oc-1* and *boule* RNAi treatments.

5. In page 10, the authors explore the similarity of epistatic interactions between *oc-1* and *boule* with *nanos-1* and *eled*. While it is clear that both genes show similar epistatic interactions with these genes, the nature of these epistatic relation is not clearly shown. The authors should include single *nanos-1* and *eled* RNAi experiments and include the quantification of Edu⁺ cells in testes in both cases. Additionally, if they want to explore the nature of these epistatic relations (*oc-1/boule* with *nanos-1* and *oc-1/boule* with *eled*), the authors should include data showing similar amount of gene downregulation in both single and double RNAi experiments. Finally, further description of the double RNAi experiments should be included in the methods section.

As the reviewer suggested to improve comparisons between single and double knockdowns, we have now included single *nanos-1* and *eled* RNAi results in **Fig. 5b**, and associated quantification in **Fig. 5c**. Please note that *nanos-1* RNAi, *nanos-1/oc-1* RNAi, and *nanos-1/boule* RNAi worms had too few germ cell nuclei to properly quantify statistics and thus are omitted in **Fig. 5c**. We have also validated the knockdown of target genes in both single and double RNAi, which is now reported in the new **Supplementary Fig. 8**. In addition, we have included more experimental details about double RNAi experiments in the methods section (**lines 396-408**).

Minor comments:

1. The authors should describe in detail the number of genes that were used to calculate the principal components used to build the UMAP plot.

As now clarified in the methods section (**lines 377-380**), 3,000 genes were used to calculate the PCs of the full dataset, and 8,000 genes were used to calculate the PCs of the stem cell population. These numbers were chosen by SAM based on the size of the datasets to satisfy computational memory constraints of the algorithm.

2. There is a typo in FigS1 legend “FISH image. es: esophagus.”

Fixed. Thanks.

3. The authors should provide quantification of EdU⁺ cells of all confocal images included in the paper.

The quantification of EdU⁺ cells of all confocal images in main figures is reported in **Fig. 3c**, **Fig. 4c**, and **Fig. 5c**. The only exceptions are *nanos-1* RNAi experiments in which germ cells

are too few to properly quantify.

Reviewer #2

Major comments:

*1. Does *oc-1* and *boule* inhibition cause over proliferation or arrested proliferation? It is interesting that these treatments increase EdU staining but do not appear to cause enlargement of that organ. Could *oc-1* and *boule* promote transit through a specific cell cycle stage, such that their inhibition leads to higher EdU uptake but lower output of proliferation and differentiation? A complementary analysis, for example with H3P or FACS, would be helpful to resolve this.*

We thank the reviewer for bringing this important issue to our attention. While the enlargement of testes was not observed, GSCs were packed at higher number densities after *oc-1* RNAi and *boule* RNAi. This new important result is consistent with GSC over-proliferation and now presented in **Fig. 3d**. In addition, we have observed apoptosis in GSCs (shown in the new **Supplementary Fig. 5**), which was not affected by *oc-1* RNAi and *boule* RNAi, suggesting that schistosome GSCs undergo continuous turnover and need to actively divide to compensate for this constant loss.

We now also do a better job in distinguishing the two concepts: over-proliferation and faster cell division. Over-proliferation is the outcome of GSCs undergoing more mitotic divisions to produce more daughter GSCs, which does not necessarily indicate more divisions per unit time. Our EdU experiments suggest that more GSCs are active in the process of DNA synthesis during a given time period after *oc-1* RNAi and *boule* RNAi, but does not directly measure the rate of cell division. We have now revised the text to clarify this important point (**lines 142-154**).

H3P labels not only mitotic cells but also cells that are in meiosis as histone H3 phosphorylation correlates with chromosome condensation (Hans & Dimitrov, *Oncogene*, 2001, 20:3021-3027). Therefore, the comparison between control and RNAi experiments is unjustified as germ cell differentiation/meiosis was drastically reduced after *oc-1* RNAi and *boule* RNAi. On the other hand, although FACS is a widely used method to determine cell cycle progression, our previous study showed that the majority of cycling cells in schistosome juveniles are somatic stem cells (**ref. 5**), this unfortunately does not allow us to quantify the GSC cell cycle using FACS.

*2. The aspect of the *oc-1* and *boule* phenotypes involving loss of differentiation should be more strongly supported either with markers of spermatids or quantifications of the DAPI morphological cell type scoring.*

We have screened schistosome homologs of known planarian spermatid markers and searched carefully for genes specifically expressed in spermatids in the recent schistosome adult cell atlas (**ref. 43**), but failed to identify a marker gene that specifically labels spermatids. However, as the reviewer suggested, our new **Fig. 3e** now shows drastic reduction in the number of differentiated germ cells in worms after *oc-1* RNAi and *boule* RNAi treatments using DAPI morphological cell type scoring. This is further corroborated by the new **Fig. 2d-e and Fig. 3f-g**, which reveals broader *nanos-1* and *eled* expression, both of which are GSC markers and excluded

from differentiated germ cells, after *oc-1* RNAi and *boule* RNAi compared to control RNAi experiments. Taken together, the reduced differentiation phenotype is now better characterized by this new set of data and analyses.

3. *The data presented in the double RNAi experiment (Fig 5b and c) is incomplete because there are quantifications of conditions (such as oc-1 RNAi alone or boule RNAi alone) that appear to be missing visualizations in 5b.*

Fixed. Results from the single RNAi experiments are now shown in **Fig. 5b-c**.

4. *How were the double RNAi experiments conducted? In other systems, simultaneous treatments with two different dsRNAs can reduce the effects of inhibition from either, leading to the need to have comparative single-gene inhibition conditions dosed with competing control dsRNA in tests of gene interaction. For example, the penetrance or expressivity of animals treated with oc1+control dsRNA is likely to be lower than that of oc1 dsRNA alone, which could impact the interpretations of the study. In that case, low EdU incorporation in the oc1+elad dsRNA condition could either be due to genetic interaction as authors conclude or due to competition between the oc1 and elad dsRNAs. There would be a similar issue with the other double RNAi experiments. I'm concerned that without the aforementioned design or some complementary analysis (such as verifying knockdown is equivalent in the single- versus double-RNAi) then it is less certain that the results reveal a genetic interaction.*

We thank the reviewer to raise this important point. We now have validated efficient gene knockdown in double RNAi experiments, as shown in the new **Supplementary Fig. 8**. Unlike the original manuscript where we had taken this point for granted, we now acknowledge this explicitly in the text (**lines 203-206**). In addition, we have provided more experimental details of double RNAi in the methods section (**lines 396-408**).

Minor comments:

1. *How does oc-1 and boule expressions look like on the tSNE plots as in Fig S1?*

We thank the reviewer for this great suggestion. We now show UMAP projections overlaid with *oc-1* and *boule* expression in **Fig. 1d**. While *oc-1* is expressed throughout the *nanos-1*⁺ cluster, *boule* and *nr* are expressed in two subclusters. Since *nr* is known to be expressed in the *nanos-1*⁺ stem cells (also known as S1 cells) in primordial vitellaria, these observations suggest that the *nanos-1*⁺ cluster contains two transcriptionally similar but distinct cell types: S1 cells (*nr*⁺) and GSCs (*boule*⁺). We have revised the text (**lines 119-126**) to clarify this important point.

2. *Only some of the stages are marked in Fig 3b. If some are shown for the knockdown conditions, it would be helpful to present them for the controls as well.*

Fixed. Thanks.

3. *Can stages of PGC differentiation be identified through the scRNAseq analysis, for example via trajectory analysis? How do these correspond with states of oc-1 and boule expression?*

Unfortunately, our dataset does not contain sufficient numbers of GSCs and differentiating/differentiated germ cells for differentiation trajectory analysis.

4. *The differences in the role of oc-1 between S. mediterranea and S. mansoni are intriguing. Could they be due to differences in the knockdown efficiency in these two organisms?*

Both early and late knockdown of *Smed-oc-1* led to phenotypes that suggest *Smed-oc-1* is essential for GSC maintenance, which is completely different from the function of *Sm-oc-1*. In addition, **Fig. 6e** and the new **Supplementary Fig. 8** suggest that *oc-1* homologs were efficiently knocked down to the level below the WISH detection limit in both *S. mediterranea* and *S. mansoni*. These converging lines of evidence allowed us to conclude that the different phenotypes of *oc-1* RNAi is unlikely caused by the difference in RNAi knockdown efficiency.

Reviewer #3

Major comments:

1. *Same protocol including cell isolation and Smart-Seq2 program was used for current study and previous studies (ref 5, 25). Please highlight how the author made a progress and they were able to detect of lowly expressed genes which were undetectable in previous study (ref 25). Are there any parameters indicate those results generated by Smart-Seq2 from lowly expressed genes are reliable data?*

We apologize for the confusion. As now clarified in the text (**lines 92-95**), our previous studies (**ref. 5 and 25**) sequenced flow-sorted G₂/M phase cells to enrich for stem cells that are undergoing mitosis. While this approach allowed us to identify μ , δ' , ε_α , ε_β somatic stem cell populations, we noticed that the data lacked GSCs, indicating that some stem cell types were insufficiently sampled using our previous flow-sorting strategy. Therefore, in this work, we performed scRNAseq on all body cells, which has allowed us to identify new stem cell types, including GSCs. We do not claim to detect more lowly expressed genes compared to the previous work.

2. *In this study, RNAi was undertaken by targeting different genes in parasites. To evaluate knockdown efficacy of RNAi, we normally use real time PCR to measure the transcription level of the targeted gene or identify phenotype changes of the targeted protein. Both of those two methods are important to verify the knockdown of targeted gene to ensure the difference observed in the imaging (in situ hybridization or Edu staining) was not induced by off-targeted genes, which is normally presented in dsRNA knockdown system in schistosomes.*

We have now validated the gene knockdown in both single and double RNAi using WISH, as reported in the new **Supplementary Fig. 8**. As the number of GSCs changed significantly after *nanos-1*, *oc-1*, and *boule* RNAi treatments, WISH is better suited to confirm knockdown compared to real time PCR, as RT-PCR measures the total expression level which depends on both the number of cells expressing the gene of interest and expression levels of that gene in individual expressing cells.

We also ruled out potential off-target effect by using dsRNAs that target distinct regions of the same gene (an example is shown in **Supplementary Fig. 10**). Our observation is consistent with a recent genome-wide RNAi screen that also observed minimal off-target effect in dsRNA induced gene silencing in *S. mansoni* (Wang et al. Science, 2020, 369:1649-1653).

3. The same location of oc-1 and nanos-1+ does not mean there are interaction between those two genes. Furthermore the location of nanos-1+ cells in the yolk cell-producing somatic reproductive organ was performed previously (ref 46), not undertaken with the oc-1 in the same experiment, which is hard to compare. Similarly, co-expression of oc-1 and boule in the spermatogonial cells indicated potential interaction presented between those two cells. However, to determine if there is the direct or indirect regulation of oc-1 and boule, more protein functional analysis and protein interaction assays are needed.

Thanks for bringing to our attention this potential point of confusion, which we do our best to clarify in the revised text. We agree with the reviewer that co-localization/co-expression does not indicate regulation or interaction, and indeed none of our analysis of genetic interactions was based on co-localization. Instead, these epistatic relations were defined by how the phenotypic effect of a gene is dependent on or influenced by the presence or absence of another gene. For example, we showed that *oc-1* RNAi phenotype was dependent on the presence of *nanos-1* suggesting that *nanos-1* functions upstream of *oc-1* in GSCs, but this relationship is not directly transferrable to primordial vitellaria cells as the phenotypic relation was observed in GSCs. This approach and associated terminology have been used in many genetic studies to reveal genetic relations between genes, which are sometime consistent with but not necessarily always equivalent to biochemical interactions.

Minor comments:

1. I suppose that Fig2d (2e) only showed the expression of nanos-1 and eled after oc-1 RNAi. Please show the control cells without oc-1 RNAi as well. Same issue with Fig 3d and 3e.

Fixed. The control samples are now shown in **Fig. 2d-e** and **Fig. 3f-g** (previous Fig. 3d-e).

2. This study indicated the roles of oc-1 and boule in suppressing proliferating and promoting differentiation. Can you clarify the relationship between proliferation and differentiation and provide evidences or refs to support it?

As now clarified in the text (**lines 253-256**), male GSCs have the ability to proliferate through mitotic divisions to produce more GSCs (self-renewal) and differentiate into spermatocytes that progress through meiosis to produce spermatids and sperm. These are two essential functions of the GSCs, which have been reviewed by many. One excellent example can be found in Spradling et al. Cold Spring Harb. Perspect. Biol., 2011, 3:a002642.

Reviewers' Comments:

Reviewer #1:

Remarks to the Author:

In this revised version of the manuscript, Li et al. present an improved version of the manuscript, providing more details on the computational methods used and additional figures to support the previous claims. Altogether, the authors have answered to all my questions and critics. I have no further comments

Reviewer #2:

Remarks to the Author:

These revisions address all of my comments completely and I congratulate the authors on a very elegant study.

Reviewer #3:

Remarks to the Author:

My questions have been addressed.